# Can New-Type Urbanization Promote Enterprise Green Technology Innovation?—A Study Based on Difference-in-Differences Model

**Ran Zhang [1], Guoquan Kong [1,*] and Huaping Sun [2]**

1     School of Economics, Qingdao University, Qingdao 266061, China
2     School of Finance and Economics, Jiangsu University, Zhenjiang 212013, China
*     Correspondence: 425248807kgq@gmail.com

**Abstract:** China proposed a new-type urbanization (NTU) strategy in 2012 to solve ecological and environmental problems caused by the traditional rapid and rough urbanization development model. Focusing on the policy's important goal of building green and smart cities, it is crucial to explore whether the pilot of NTU promotes green innovation at the enterprise level, and thus achieves green environmental protection. Based on data from 1717 Chinese listed companies' green patent applications between 2011 and 2020, this paper studies the impact effect of NTU on enterprises' green technology innovation utilizing the difference-in-difference model combined with the PSM-DID method. The findings indicate that: NTU has a substantial effect on enterprise innovation in green technologies. The mechanism analysis shows that NTU can encourage green technology innovation in enterprises by easing their financial restrictions. This requires the government to encourage enterprises to engage in green technology innovation by alleviating their financing constraints and reducing their debt financing costs through policy incentives and financial subsidies. Heterogeneity analysis shows that the impact of the policy on green innovation is more significant in the central and western regions, highly marketable areas, non-heavy-polluting industries, and among enterprises with higher levels of green innovation.

**Keywords:** new-type urbanization; green technology innovation; financing constraints; debt financing costs

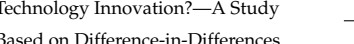



## 1. Introduction

Environmental pollution has always been a global issue that has received widespread attention. In the long-term traditional urbanization process, the agglomeration of industrial production and energy consumption [1,2] has generated problems, such as excessive carbon emissions, excessive resource consumption, and serious environmental pollution [3], which have adversely affected the improvement of the quality and level of urbanization. To promote the transformation of environmentally friendly urbanization development, the Chinese government first proposed the concept of "NTU (new-type urbanization)" in 2012 and explicitly proposed "adopting a NTU strategy of intensification, intelligence, and green and low-carbon" at the Central Economic Conference [4]. In 2014, "The National NTU Plan (2014–2020)" (hereinafter referred to as "the plan"), published by the Chinese central government, revealed a new urbanization path, emphasizing that the core of NTU is people-oriented [5,6].

Green technology innovation specifically focuses on synergistic development, which is in line with NTU's focus on balancing the relationship between the economy and the environment. Innovation in green technology can boost the effectiveness of resource usage and lower pollution emissions during production [7,8], thus helping to achieve coordinated development of the economy and the environment [9]. According to studies [10,11], green technology innovation is the primary strategy for addressing the challenges of global

environmental contamination. Green technology innovation can not only help urban economies to achieve low-carbon sustainable growth, but also, improve urban economic efficiency [12]. Green technology innovation in enterprises is an important way to achieve economic and environmental "Win-win" results. The NTU policy emphasizes that China should integrate the concept of ecological civilization into the processes of urbanization and green industry at the same time; new urbanization construction also encourages enterprises to clean production and apply green technology innovations. Therefore, the NTU policy is likely to promote green technology innovations in enterprises.

The data from 1717 listed companies in China from 2011 to 2020 are used to investigate how NTU's transmission mechanism affects the development of green technology in enterprises. Compared with previous studies, the contributions are as follows: (1) The existing literature has not yet studied the implementation effects of NTU in the context of corporate green technology innovation. In this paper, we investigate the NTU policy from the perspective of green innovation and propose and verify the effect of the policy on the impact of green technology innovation in enterprises. (2) The majority of NTU's research has utilized data from provincial and local levels to examine and study the policy. Because enterprises are the primary source of innovation in green technology, this paper utilizes more precise patent data to quantify enterprise innovation and progresses from the macro level to the micro level. (3) Since China has not yet published a clear indexing system for evaluating new urbanization, the portrayal of new urbanization is prone to reverse causality problems when they are measured using specific data. The use of a difference-in-differences model can effectively avoid the endogeneity problem among variables.

The rest of this paper is structured as follows: The literature review is presented in Section 2. Section 3 contains the theoretical analysis and research hypotheses. The data and empirical methodologies are introduced in Section 4. Section 5 displays the empirical outcomes of regression models. The conclusion and policy recommendations are covered in Section 6.

## 2. Literature Review

NTU is a comprehensive policy at the city level to realize the transformation of the traditional urbanization model of "factor-dependent" and "investment-driven" data. The policy includes environmental regulation policies, proposes to integrate ecology into the entire urban development process, and emphasizes the path of intensification, intelligence, green processes, and low carbonization. The policy is characterized by weak constraints, industry specificity, and policy combination. The literature review is undertaken from three perspectives of NTU and enterprises' green technology innovations based on the research material.

### 2.1. Research on NTU

In optimization of the energy consumption structure, NTU has a more significant energy-saving effect in resource-rich areas [13], which has a certain inhibitory effect on energy efficiency initially [14], but the inhibitory effect diminishes with the increase in the urbanization level [15]. Studies have shown that the structure of energy consumption is only influenced by economic development [16]. As the level of urbanization continues to increase, advances in energy production and consumption technologies have promoted industrial transformation and upgrades and clean energy consumption [17,18], thus improving energy efficiency. In addition, some scholars have calculated that a 1.0% increase in the urbanization level will result in a corresponding decrease in the energy consumption level of 0.089%, and the surrounding area will benefit from the beneficial spillover effect [19].

In terms of reducing carbon emissions and air pollution during the transition to NTU, due to the absence of the completion of energy upgrades, carbon dioxide emissions will rise along with the rise in industrial and household energy use levels [20,21]. However, with the increasing level of urbanization, long-term carbon emissions can be decreased

by NTU by increasing the amount of clean energy used [22]. Zhao and Wang [23] found that the effect of NTU on air pollution was regionally heterogeneous based on the spatial Durbin model and spatial mediation model, significantly reducing carbon emissions and the amount of local air pollution.

In terms of intensive land use, Zhao et al. [24] used the super-efficient DEA model and Malmquist index analysis to conclude that the relationship between land eco-efficiency and NTU exhibits an N-shaped curve. Zhang et al. [25] found a significant non-linear relationship between land transfer and NTU based on a threshold panel model approach. However, the application of NTU improved intensive land usage and land eco-efficiency over time [26].

### 2.2. Research on Enterprises' Green Technology Innovation

Enterprises' green technology innovation is influenced by both internal and external influences, which have been thoroughly researched from various perspectives. The authors of [27] identified organizational aspects within enterprises, such as enterprise environmentalism, green management systems, and green intellectual capital, as the foundation for green technology innovation in enterprises. Several academics have suggested that R&D investment covering both capital and personnel and legitimacy pressures from stakeholders is highly correlated with enterprises' ability to produce innovative green technologies [28,29].

In the Chinese context, most research perspectives on external factors affecting enterprises' green technology innovation focus on institutional pressure or support. Several academics have found that government financing, tax breaks, and R&D subsidies significantly boost enterprises' green technology innovation [30,31]. The ETS is effective in fostering green innovation, and the impact is amplified in less competitive areas [32].

### 2.3. Research on the Impact of NTU on Enterprises' Green Technology Innovation

The research on NTU's effects on green technology innovation in enterprises is within the bounds of how environmental control policy impacts this field. Previous studies show that NTU has strengthened the environmental regulation of pilot cities.

Environmental regulation policies refer to the government's implementation of appropriate incentives and constraints for enterprises to fulfill the goals of energy conservation, emissions reduction, and environmental enhancement. Most scholars believe that an appropriate level of environmental regulation will make the "innovation compensation effect" greater than the "compliance cost effect" [33,34], thereby forcing enterprises to engage in environmentally friendly technology innovation, while greatly enhancing their capacity for such innovation [14]. Some scholars have argued the opposite, arguing that increased environmental regulations will force enterprises to pay additional production and operating costs [35,36], leading to poorer business performance, crowding out investment in R&D, and discouraging technological innovation [37,38].

The abovementioned studies have analyzed and explored different perspectives on NTU that have been carried out, but there are still limitations in the following aspects: (1) Although some of the literature evaluated the effects of NTU using traditional policy evaluation methods, most of the selected research studies are focused on energy consumption, carbon emission, and intensive land use, and there are no studies that look into NTU implementation's consequences from the standpoint of innovation. (2) Although the construction of a comprehensive index system to evaluate the urbanization development of cities can cover different dimensions and levels of urbanization, the selection of indicators and the weights assigned to them are subjective, and therefore, it is impossible to measure the effects of NTU from a unified perspective. (3) Although the literature is rich in tests of Porter's hypothesis, the studies mainly focus on a single environmental policy, with limited tests on a comprehensive policy such as NTU.

## 3. Theoretical Analysis and Hypotheses

### 3.1. The Effect of NTU on Enterprises' Green Technology Innovation

The implementation of NTU will bring about a change in the development goals of cities to form a quality-centered and inclusive growth model. Technological innovation by enterprises will unavoidably accompany this process, which will lead to the upgrading of current technologies and the creation of eco-friendly technologies that meet the criteria for low-carbon development, thus realizing "the Porter Hypothesis". According to "the Porter hypothesis," there are two ways that environmental legislation influences enterprises' production choices. Environmental regulations, on the one hand, have a cost of compliance effect, which increases the costs of emission reduction and pollution control. This may lead enterprises to cut back on R&D investments in the short run or a shift to other types of investments. On the other hand, a well-designed environmental policy can have an innovation compensation effect [39]. After realizing the long-term nature of environmental regulations, enterprises will invest in abatement technologies in advance to achieve the emission levels required by regulations, ultimately achieving both environmental and economic benefits [40,41]. Compared with environmental legislation, although the NTU policy does not directly carry out environmental legislation, it can form a "soft constraint" on the green development of enterprises; this constraint is reflected in the development of the new type of urbanization that requires a new type of development to drive the urban economy to avoid high energy consumption and high levels of emissions of development model to achieve a green low-carbon industrial economy transition, and then produce environmental effects; this shows that the NTU policy is a "soft constraint" environmental regulation policy. Based on this, the relationship between NTU and enterprises' green technology innovation can be analyzed with the help of "the Porter Hypothesis". On the one hand, enterprise is a key body of industry development, NTU is a "soft constraint" environmental regulation policy, and companies are required to drive technological change through innovation to promote green production through green technology innovation that caters to low-carbon development; on the other hand, "the Porter Hypothesis" suggests that environmental legislation will increase the companies' willingness to innovate in green technologies, while the NTU policy's "soft constraint" has a similar effect. The environmental costs of companies are offset by the compensation effect of innovation, providing conditions for companies to create innovative sustainable green technologies and enabling companies to take on social responsibility and enhance their competitiveness. Accordingly, Hypothesis H1 is proposed.

**Hypothesis 1.** *The implementation of NTU may promote enterprises' green technology innovation.*

### 3.2. Heterogeneity Analysis of NTU

The influence on NTU's promotion of green technology innovation may also vary depending on where the enterprises are located, the degree of marketization, the difference in industry pollution levels, and the level of enterprises' green technology innovation.

Most pilot regions have introduced a series of mandatory policy requirements for high-energy-consuming and high-emission industries, implementing stricter entry standards for new projects in high-carbon-emission and high-polluting industries, while a series of incentive-based policies have been introduced for clean, low-carbon industries [42]. Most enterprises in central and western China are traditional industries that have followed a high-energy-consumption and high-emission R&D and production route in the past, while clean, low-carbon industries are mostly concentrated in the eastern coastal regions. While enterprises in the eastern areas have reduced the regulatory stress and costs, those in the central and western regions are subject to stricter policy requirements [43,44].

From the perspective of the marketization degree, the marketization degree is an important factor affecting the effect of NTU policy implementation. On the one hand, the higher the degree of marketization is, the higher the degree of attention to the business environment is, and the higher the concern for the ecological environment is. On the other

hand, enterprises in regions with a stronger degree of marketization face more intense competition. Enterprises need to reduce pollution reduction costs through green technology innovation to meet environmental requirements. At the same time, they can also experience the technology spillover effect to reduce emissions, thus having the market competitive advantage [45]. Therefore, the higher the degree of marketization of the region is, the higher the enterprise's green technology innovation momentum is.

At present, China's economy is in a period of structural transformation, and there are great differences in the pressures of market competition, the levels of technological development, and the demand for green innovation among different industries; in particular, heavy- and non-heavy-polluting industries employ green technology innovation differently. On the one hand, enterprises in non-heavy-polluting industries are more capable of sustainable development and can obtain more external support resources, which can help to promote enterprises' green technology innovation. For example, the Chinese industrial policy focuses on supporting the production of high-tech goods and equipment and the construction of social and public facilities and strictly limits the production of goods that are obsolete, waste resources, and pollute the environment [46]. On the other hand, heavy-polluting enterprises face more direct pressure to deal with pollution, requiring more human, material, and financial resources to invest in direct pollution control, resulting in the crowding out of resources for green technology innovation; therefore, it is not conducive to green technology innovation.

In addition, at the level of enterprise innovation capacity, under policy pressure, enterprises with a high innovation capacity can allocate a large number of resources more actively and efficiently to promote green technology innovation by enterprises. This is because in terms of building an innovation talent pool and a low-carbon technological R&D base, they are more advantageous. In contrast, low-innovation-capability enterprises have fewer options due to the limitations of their own development model, and it is difficult for them to make changes in a short period of time [47]. Accordingly, these hypotheses are proposed:

**Hypothesis H2a.** *Compared with the eastern region, NTU is more helpful for promoting green technology innovation by enterprises in China's central and western prefectures.*

**Hypothesis H2b.** *Compared with those in low-marketization regions, NTU is more helpful in promoting enterprises' green technology innovation in high-marketization regions.*

**Hypothesis H2c.** *Compared with the enterprises in heavy-polluting industries, NTU is more helpful for promoting green technology innovation by non-heavy-polluting industries.*

**Hypothesis H2d.** *Compared to low-innovation-capacity enterprises, NTU is more helpful for promoting green technology innovation by high innovation capacity enterprises.*

*3.3. Mechanistic Analysis of NTU*

According to Schumpeter's innovation theory, financing is a key factor in technical innovation. Due to return uncertainty, information asymmetry in the innovation process, and greater regulatory expenses, creative operations are vulnerable to severe external financial limitations, and financial constraints stifle enterprises' innovative efforts [48]. Since green technology innovation differs from traditional technology innovation in that it requires more initial capital investment, a longer profit cycle, and unpredictable risks, it requires additional financial support to address market failure issues such as environmental externalities, path dependence, and a flawed capital market. This suggests that additional funding is required for green technology innovation to achieve step change innovation [49,50], and consequently, enterprises may be more prone to having financial limitations when they are developing green technologies. Those organizing low-carbon city planning initiatives in pilot zones have suggested green finance policies, including tax breaks, industry-specific

subsidies, loans with advantageous interest rates, and specialist funds for low-carbon growth. These green financial policies can reduce investments in polluting projects and provide more funds for green industries and eco-friendly manufacturing techniques, relieving any financial restraints that enterprises may experience and promoting the advancement of green technical innovation. Accordingly, Hypothesis H3 is proposed.

**Hypothesis H3.** *NTU can promote enterprises' green technological innovation by alleviating enterprise financing constraints.*

## 4. Methods and Data

### 4.1. Data Sources

The listed companies selected in this paper are those whose listed companies' offices are mainly located in cities. Thereby, the number of listed companies in cities where they are located represents the level of green innovation in that city or region. From 2011 to 2020, patent data and corresponding enterprise-level economic data of A-share listed companies in China's Shanghai and Shenzhen stock markets were selected for this paper. Among them, data on enterprise microeconomic characteristics were obtained from the China Stock Market and Accounting Research Database (CSMAR). As for the research samples, they were evaluated based on the following standards: (1) the samples of ST-, *ST-, and S*ST-listed companies with abnormal financial status were removed; (2) the research samples with missing variable index data were removed; (3) due to the manufacturing sector's dominance of green technology innovation and the unique characteristics of banking and real estate sectors, it is difficult to measure innovation performance. Therefore, the research samples of service industries such as finance and real estate were excluded from the screening.

Chinese Research Data Services (CNRDS) provided the main green patent data for this paper, and the specific analysis methods are as follows: The "Green List of International Patent Classification" introduced by the World Intellectual Property Organization (WIPO) in 2010, combined with the international patent classification number, were used for the screening of green patents of listed companies in order to screen and extract green patent data from the sample's listed companies. Combining the worldwide patent classification codes enables the extraction of the green patent information from the sample's listed companies. Green utility model patents and green inventive patents are two further types that highlight the different creativity and value of green inventions, and it is generally believed that the former one is more innovative than the latter one is.

### 4.2. Indicator Construction

Since we focus on the impact of NTU policies on green technology innovation before and after implementation, NTU is regarded as a quasi-natural experiment; the explanatory factor is the quantity of green patent applications submitted by listed companies because the research perspective is to assess NTU's effectiveness through the activities of listed companies that produce innovative green technology [51]. The reasons for this are as follows: First, with quantifiability and spillover effects inside and beyond the business, the outcomes of business operations for green technology innovation are most naturally reflected in green patents [52]. Second, considering that the patent application process is time-consuming, patent application data rather than patent grant data can be used to examine enterprises' green technology innovation efforts and the effects of pilot programs in a more time-sensitive way [53]. The dataset of green patent applications submitted by listed companies in the current year was processed logarithmically. Green patent applications overall (gpa), green invention-based patent applications (gipa), and green utility model applications (gupa) are three different kinds of green patent applications.

Dummy variables for policy grouping were set based on whether the cities were included in the NTU list, dummy variables for time grouping were set based on the release time of the pilot list, and the interaction term of both was used as the core explanatory variable. The first batch of policy pilot areas is composed of 62 cities (towns) in Jiangsu and Anhui provinces and Ningbo; the second batch of pilot areas is composed of 59 cities (towns) in Fangshan District in Beijing and 14 pilot areas undergoing rural land system reform; the third batch of pilot areas is composed of 111 cities (towns) in Shunyi District in Beijing, etc. Meanwhile, for the convenience of measurement, pilot areas are unified according to the lowest unit of the prefecture-level city to which they belong in this paper. In addition, considering the lag between the policy and the release time of each batch of pilot cities list, this paper uses 2014, 2015, and 2016 as the pilot policy's time nodes.

Due to the potential impact that additional factors at the enterprise and city levels may have on enterprises' green technology innovation, a variety of enterprise economic characteristics and city-level influencing factors were selected as control variables in this paper. (1) Enterprise size (*ln*size): In the literature, enterprise size has been demonstrated to have a significant impact on enterprise innovation [54]. The logarithm of the total capital at year end is used to determine the size of the business in this paper. (2) Enterprise age (*ln*age): An enterprise's age typically indicates how mature it is, and mature enterprises are more likely to be innovative. Here, the age of the enterprise is measured by the logarithm of the length of time the enterprise has been listed. (3) Enterprise indebtedness (lev): Enterprise indebtedness shows how the market views an enterprise's creditworthiness, and a moderately indebted operation gives enterprises more resources for cutting-edge initiatives such as modernizing their technology infrastructure and operational procedures. Enterprise indebtedness is calculated using the logarithm of the ratio of the enterprises' current year loan amount to total assets. (4) Variables related to enterprise performance and governance structure [55,56]. I decided to simultaneously control for enterprise return on total assets (roa), the shareholding ratio of the top shareholder (*ln*top1), and the type of enterprise ownership (soe) in order to prevent the impact of factors such as enterprise performance and governance structure on enterprises' green technology innovation. Among them, the ratio of an enterprise's total assets to its net profit is known as the roa; the top1 shareholding ratio is the ratio of the shareholding share of the first largest shareholder to total equity; for the nature of enterprise ownership, state-owned enterprises take the value of 1 and private enterprises take the value of 0. Table 1 displays descriptive statistics for each primary variable, and the full sample contains 1717 enterprises with 17,086 samples in the panel data.

*4.3. Econometric Model Construction*

The DID method was used to divide the subjects into a treatment group (the area where the policy was implemented) and a control group (the area where the policy was not implemented). The overall impact of the policy was calculated by comparing the trends over time before and after the policy was implemented, and whether the policy was implemented or not in the treatment group and the control group was compared to exclude other factors that changed over time and were not important. This paper divides the study subjects into treatment (regions that implemented the policy) and control (regions that did not implement the policy) groups using the DID (difference-in-differences model) method. By contrasting the policy's trend over time before and after implementation, I then calculated the overall impact of the policy, as well as compared whether the policy was implemented in the treatment and control groups, in order to rule out other factors that change over time and are unimportant.

**Table 1.** Descriptive statistical characteristics of the main variables.

| Variables | Indicator Meaning | Whole Sample | | | | Treatment Group | | | | Control Group | | | |
|---|---|---|---|---|---|---|---|---|---|---|---|---|---|
| | | Average Value | Standard Deviation | Min | Max | Average Value | Standard Deviation | Min | Max | Average Value | Standard Deviation | Min | Max |
| gpa | Logarithm of Green Patent Applications | 0.950 | 1.236 | 0 | 7.319 | 0.618 | 0.985 | 0 | 6.471 | 0.685 | 1.086 | 0 | 6.820 |
| gipa | Logarithm of Green Invention Patent Applications | 0.648 | 1.032 | 0 | 6.820 | 0.915 | 1.190 | 0 | 6.477 | 0.993 | 1.286 | 0 | 7.319 |
| *ln*age | Logarithm of enterprise age | 2.778 | 0.404 | 0 | 3.807 | 2.788 | 0.398 | 0 | 3.664 | 2.767 | 0.411 | 0.693 | 3.807 |
| soe | Nature of enterprise equity | 0.425 | 0.494 | 0 | 1 | 0.415 | 0.493 | 0 | 1 | 0.438 | 0.496 | 0 | 1 |
| *ln*size | Log of total enterprise assets | 3.984 | 1.285 | 0.993 | 10.220 | 3.948 | 1.190 | 0.993 | 8.202 | 4.029 | 1.390 | 1.075 | 10.216 |
| lev | Enterprise gearing | 0.423 | 0.200 | 0.007 | 1.150 | 0.428 | 1.198 | 0.007 | 1.056 | 0.417 | 0.202 | 0.008 | 1.150 |
| *ln*top1 | Logarithm of the shareholding ratio of the largest shareholder | 3.427 | 0.483 | −1.238 | 4.500 | 3.428 | 0.471 | −1.238 | 4.500 | 3.427 | 0.498 | 1.099 | 4.458 |
| roa | Net profit margin on total assets | 0.037 | 0.065 | −1.057 | 0.590 | 0.037 | 0.062 | −0.847 | 0.526 | 0.037 | 0.068 | −1.057 | 0.590 |
| cost | Cost of debt financing | 0.004 | 0.065 | −2.455 | 0.947 | 0.008 | 0.046 | −0.797 | 0.164 | 0 | 0.083 | −2.455 | 0.947 |
| kz | kz index | 1.049 | 2.257 | −11.340 | 10.230 | 1.053 | 2.238 | −11.256 | 8.987 | 1.045 | 2.280 | −11.344 | 10.234 |
| number | | 17,086 | | | | 9417 | | | | 7669 | | | |

At the same time, the assumption that the experimental group and the control group meet the common trend is the prerequisite for the unbiased results of the double difference estimation; in this paper, before using DID method, I needed to make the experimental group and control group as similar as possible in all aspects of the characteristics, which involved selecting samples for the control group that were as similar as possible to the experimental group characteristics. In order to solve this problem, this paper first adopted the propensity score matching (PSM) method to eliminate the selection bias of samples, and then combined it with the DID method to solve the endogenous problem. The three batches of provinces and cities that were covered by NTU pilot projects were placed in the treatment group, and the remaining provinces and cities were -placed in the control group in order to statistically assess NTU. The specific model settings are as follows.

$$gipa_{it} = \beta_0 + \beta_1 Treat_i \times Post_t + \beta_2 DU_{it} + \beta_3 DT_{it} + \beta_4 X_{it} + \varepsilon_{it} \tag{1}$$

In Equation (1), $gipa_{it}$ indicates the amount of green invention patents applied by a certain listed company $\rangle$ in the year $\sqcup$. $Treat_i$ denotes the dummy variable of pilot areas in which NTU was used, which takes the value of 1 when the three batches of policies declare a city or province as a pilot area, otherwise it takes the value of 0. $Post_t$ is the dummy variable before and after the policy pilot test began, which takes the value of 1 during the pilot period of NTU and 0 during the non-pilot period. The key explanatory variable is $Treat_i \times Post_t$, which represents whether enterprise $\rangle$'s city is set as a NTU pilot city at time $\sqcup$. $DU_{it}$ and $DT_{it}$ represent group dummy variables and time dummy variables, respectively. The matrix $X_{it}$ includes the first largest shareholder's (*ln*top1) shareholding ratio, enterprise size (*ln*size), enterprise age (*ln*age), enterprise liabilities (lev), return on total assets (roa), and enterprise ownership type as control variables for economic characteristics of listed companies (soe). Equation (1) lists the total quantity of enterprise green invention patent applications (gipa), and the total quantity of green patents (gpa) is also mentioned below.

In the benchmark analysis, this paper's main concern is coefficient $\beta_1$ of $Treat_i \times Post_t$. The coefficient compares the effects of NTU before and after implementation in green invention patent applications made by enterprises in the pilot region. If $\beta_1$ is markedly favorable, it means that NTU supports enterprises' green technology innovation initiatives in the pilot locations.

## 5. The Analysis of the Empirical Results

### 5.1. Analysis of the Effect of NTU on Enterprises' Green Technology Innovation

The results in Table 2 show that, at the 1% level, coefficient $\beta_1$ in column (1) is significantly positive. Coefficient $\beta_1$ in column (4) is likewise statistically positive at the 1% level after adjusting for the three fixed variables, but it has a little less influence than the former one does. This suggests that, at the overall level, to a certain extent, NTU can support business innovation in green technologies. However, after adjusting for regional and sectoral confounding factors, the significance level and magnitude of green technology innovation decline. Cheng et al. (2019) examined the effect of environmental regulatory policies on urban green total factor productivity using the PSM-DID model and found that green total factor productivity was about 2.64% higher in pilot cities than it was in non-pilot cities, and that green technological progress was the main driver. So far, Hypothesis H1 has been verified.

Regarding the control factors, it is consistent with the theoretical assumption that the type of business equity and the total amount of enterprise assets have some degree of stimulating influence on business' green technology innovation, enterprise gearing, equity concentration, and return on total assets.

**Table 2.** Impact of NTU on enterprises' green technology innovation.

| Variables | gipa | | | |
|---|---|---|---|---|
| | (1) | (2) | (3) | (4) |
| $Treat_i \times Post_t$ | 0.181 *** | 0.087 *** | 0.051 *** | 0.051 *** |
| | (10.97) | (5.67) | (3.03) | (3.03) |
| $ln$age | | −0.147 *** | 0.106 * | 0.074 |
| | | (−7.71) | (1.90) | (1.32) |
| soe | | −0.063 *** | 0.138 *** | 0.131 *** |
| | | (−3.91) | (3.51) | (3.29) |
| $ln$size | | 0.386 *** | 0.329 *** | 0.323 *** |
| | | (56.71) | (24.25) | (22.97) |
| lev | | −0.110 ** | −0.214 *** | −0.207 *** |
| | | (−2.48) | (−3.94) | (−3.77) |
| $ln$top1 | | −0.205 *** | −0.043 * | −0.053 ** |
| | | (−13.14) | (−1.74) | (−2.10) |
| roa | | −0.208 * | −0.146 | −0.170* |
| | | (−1.78) | (−1.64) | (−1.90) |
| Constant | 0.585 *** | 0.272 *** | −0.788 *** | −0.642 *** |
| | (60.10) | (3.47) | (−4.10) | (−3.31) |
| Control variables | No | Yes | Yes | Yes |
| Year FE | No | No | Yes | Yes |
| Industry FE | No | No | No | Yes |
| Id FE | No | No | Yes | Yes |
| Observations | 17,086 | 17,086 | 17,086 | 17,084 |
| $R^2$ | 0.007 | 0.204 | 0.724 | 0.728 |

Note: ***, **, and * indicate statistical significance at 1%, 5%, and 10%, respectively.

*5.2. Robustness Test*

5.2.1. Parallel Trend Test

The treatment and control groups must satisfy the parallel trend assumption in order to utilize the DID method, which guarantees the objectivity of the calculated quantity. In the benchmark model of this paper, the parallel trend assumption refers to the fact that before the implementation of NTU, enterprises in pilot cities and non-pilot cities submitted nearly identical green invention patent applications over a period of time.

Figure 1 depicts the enterprises' green technology innovation parallel trend test results, and this paper takes the fifth year before policy implementation as the benchmark group. The vertical axis is the policy dynamic effect, which is expressed as the estimated coefficient $\beta_k$, and the horizontal axis is the policy time point. From Figure 1, it can be seen that $\beta_k$ was not statistically significant prior to the implementation of NTU, indicating that there was likely no discernible difference between the green invention patent applications of enterprises in the experimental group and the control group prior to the implementation of the pilot policy. This is consistent with the parallel trend hypothesis. However, starting from the year of policy implementation, $\beta_k$ is positively significant, indicating that NTU has made progress in this area. The analysis mentioned above demonstrates that the assumption of parallel trend in this paper is supported.

5.2.2. Placebo Test

There are two common approaches to placebo testing: constructing a virtual treatment group and constructing a virtual policy implementation time.

First, I constructed a virtual processing group. The original treatment group and the control group were exchanged, and the virtual treatment group was used for DID estimation.

To address the issue of omitted variables, this paper incorporates enterprise characteristics, time, and industry fixed effects into benchmark regressions. Due to data limitations, there are still some unobservable enterprise characteristics. These unobserved trait values may have had an impact on the DID method identification hypothesis. For this reason,

I randomly selected the virtual experimental group and the control group for the placebo test, so as to randomly generate the list of pilot cities and the time of pilot policy implementation, set the pseudo-policy dummy variables, and then substituted them into the baseline model regression to extract the coefficients of the error variable $Treat_i \times Post_t$ coefficients, and then repeated the above operation 500 times. The results are shown in Figure 2.

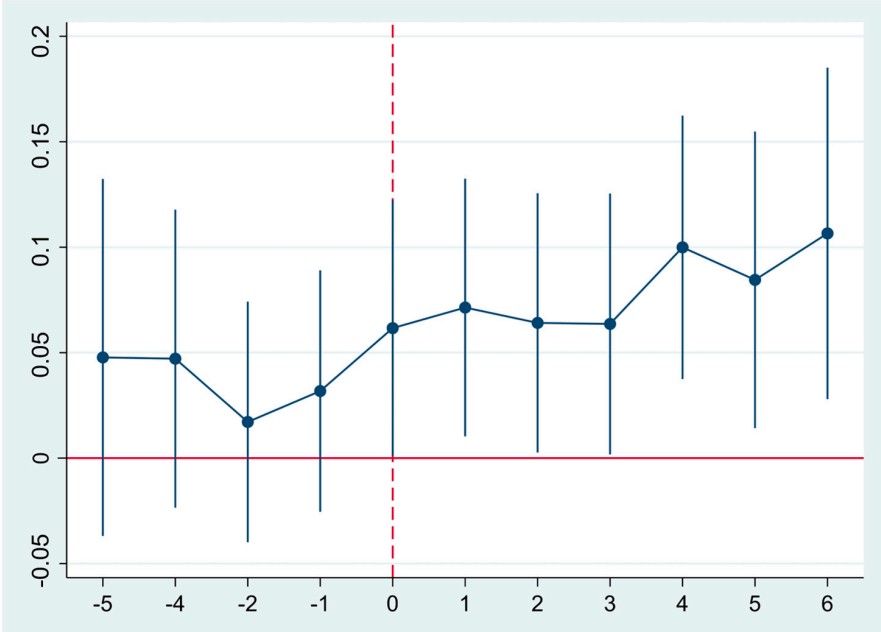

**Figure 1.** Parallel trend test.

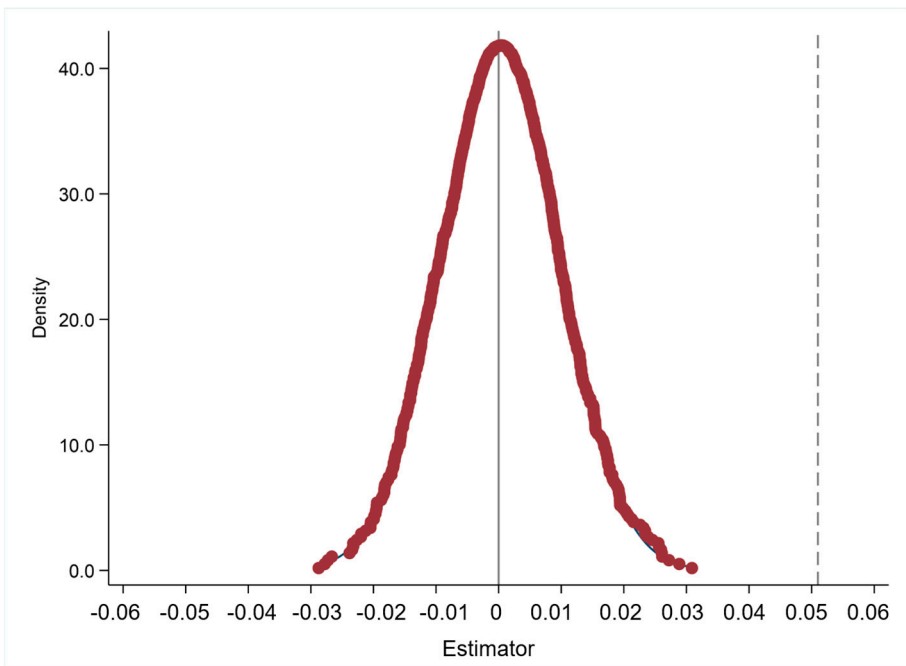

**Figure 2.** Placebo test.

This indicates that the estimated coefficients after treatment have a normal distribution and concentrate around 0, and the majority of them are not statistically significant at the 10% level, showing that other unobservable factors do not obstruct NTU's influence

on enterprise innovation in green technology and proving the accuracy of the original estimation results.

Second, I constructed the virtual policy implementation time, the implementation times of the policy of this paper are 2014, 2015, and 2016, through artificial means by assuming the policy implementation time two years ahead of schedule, for 2012, 2013, and 2014, respectively; double difference estimates were used. The results are shown in (1) in Table 3. The key explanatory variable $Treat_i \times Post_t$[1] did not have significant regression coefficients for the explained variables, indicating that the implementation time of virtual policy had no effect on the green technology innovation of enterprises; therefore, the conclusion of this paper is robust. In addition, the results of replacing the interpreted variable with the total number of green patent applications (gpa) are reported in column (2) in Table 3, and the results after excluding municipalities and provincial capitals are reported in column (3); the policy variables of NTU policy have significant positive effects on enterprises' green technology innovation, which also indicates that the research results of this paper are robust.

**Table 3.** Robustness test results.

| Variables | gipa (1) | gpa (2) | gipa (3) |
|---|---|---|---|
| $Treat_i \times Post_t$ | | 0.045 ** | 0.073 *** |
| | | (2.27) | (3.25) |
| $Treat_i \times Post_t$[1] | 0.032 | | |
| | (1.52) | | |
| $ln$age | 0.073 | 0.072 | −0.010 |
| | (1.29) | (1.10) | (−0.13) |
| soe | 0.130 *** | 0.120 *** | 0.174 *** |
| | (3.26) | (2.59) | (3.21) |
| $ln$size | 0.322 *** | 0.375 *** | 0.338 *** |
| | (22.91) | (23.01) | (17.41) |
| lev | −0.208 *** | −0.160 ** | −0.163 ** |
| | (−3.79) | (−2.51) | (−2.24) |
| $ln$top1 | −0.054 ** | −0.082 *** | −0.052 * |
| | (−2.12) | (−2.81) | (−1.66) |
| roa | −0.166 * | −0.214 ** | −0.162 |
| | (−1.85) | (−2.06) | (−1.40) |
| Constant | −0.628 *** | −0.452 ** | −0.527 ** |
| | (−3.24) | (−2.01) | (−2.11) |
| Control variables | Yes | Yes | Yes |
| Year FE | Yes | Yes | Yes |
| Industry FE | Yes | Yes | Yes |
| Id FE | Yes | Yes | Yes |
| Observations | 17,084 | 17,084 | 9207 |
| $R^2$ | 0.728 | 0.745 | 0.696 |

Note: ***, **, and * indicate statistical significance at 1%, 5%, and 10%, respectively.

### 5.2.3. PSM-DID

Before a policy is implemented, the DID method assumes that both the experimental and control groups have experienced the same trend of change. If the pilot cities of NTU focus more on supporting enterprises' green technology innovation than the other cities do, then the baseline regression results in this paper are not reliable. As a result, in order to increase the comparability of the two groups of cities and prevent selective errors in the change trends of the experimental and control groups, the PSM-DID method can be used. Specifically, the grouped dummy variables are logistically regressed using the control variables to obtain propensity matching scores, and the cities with the most similar propensity matching scores can be used as the control group. However, it is important to note that the matching equilibrium assumption must be satisfied before the PSM-DID

method is applied. This paper makes reference to the research conducted by Lmbens [57] and Guo et al. [58], in which the covariates are control variables, and the hypothesis is tested by using the "k-nearest neighbor matching method" (k = 1), with a caliper range of 0.05 for k-propensity matching. The covariate t-statistic results after matching were discovered to be non-significant. This demonstrates that the original theory, which held that there was no systematic difference between the experimental and control groups, was accepted and that upon matching, there was no notable distinction between the two urban groups. As a result, the matching treatment is effective because the difference between the standard deviation before and after matching is less than 20% in absolute terms. Upon fulfilling the aforementioned premises, PSM-DID analysis was conducted, and it was found that the regression results did not differ significantly from the baseline regression results, which proved the feasibility of the PSM-DID method and the robustness of baseline regression. Table 4 displays the specific outcomes.

**Table 4.** PSM-DID validity test and regression results.

| Variables | Matched Unmatched | Mean | | *t*-Test | | Bias (%) | gipa |
|---|---|---|---|---|---|---|---|
| | | Treated | Control | *t* | *p* > \|*t*\| | | |
| $Treat_i \times Post_t$ | | | | | | | 0.051 *** (3.03) |
| *ln*age | U | 2.7877 | 2.7668 | 3.36 | 0.001 | 5.2 | 0.051 *** |
| | M | 2.7872 | 2.791 | −0.66 | 0.509 | −0.9 | (3.03) |
| soe | U | 0.41531 | 0.438 | −2.98 | 0.003 | −4.6 | 0.076 |
| | M | 0.41562 | 0.42062 | −0.69 | 0.487 | −1.0 | (1.35) |
| *ln*size | U | 3.9484 | 4.0285 | −4.06 | 0.000 | −6.2 | 0.131 *** |
| | M | 3.9498 | 3.9238 | 1.45 | 0.147 | 2.0 | (3.29) |
| lev | U | 0.42792 | 0.41696 | 3.57 | 0.000 | 5.5 | 0.323 *** |
| | M | 0.4276 | 0.42475 | 0.98 | 0.325 | 1.4 | (22.97) |
| *ln*top1 | U | 3.4278 | 3.427 | 0.11 | 0.916 | 0.2 | −0.210 *** |
| | M | 3.4277 | 3.4196 | 1.16 | 0.248 | 1.7 | (−3.82) |
| roa | U | 0.0369 | 0.0367 | 0.20 | 0.842 | 0.3 | −0.053 ** |
| | M | 0.03693 | 0.03676 | 0.19 | 0.851 | 0.3 | (−2.08) |
| Constant | | | | | | | −0.646 *** (−3.33) |

Note: *** and ** indicate statistical significance at 1% and 5%, respectively.

*5.3. Analysis of Transmission Mechanisms*

In the process of implementing NTU, local governments have deployed green finance policies and strongly advocated financial means to support cities on the road to green and low-carbon urbanization. The implementation of green finance policies helps to enhance the convenience of enterprise financing, alleviate the financing constraints of enterprises in the process of technological transformation, and provide essential financial support for enterprises' green technological innovation.

In order to determine whether NTU can support enterprise-level green technological innovation by easing enterprise financing restrictions and lowering enterprise debt financing costs through transmission mechanism analysis, this paper adopts the kz index at the enterprise level and the debt financing cost as its points of reference.

To verify the above mechanism, this paper uses the mediating effect to further expand on the basis of Equation (1) to obtain Equation (2):

$$M_{it} = \gamma_0 + \gamma_1 Treat_i \times Post_t + \gamma_2 DU_{it} + \gamma_3 DT_{it} + \gamma_4 X_{it} + \varepsilon_{it} \qquad (2)$$

$M_{it}$ is the mediating variable, and other variables are consistent with Equation (1). According to the test procedure of mediating effect, the regression results of Equation (2) are shown in Table 5.

**Table 5.** Regression results of transmission mechanism.

| Variables | KZ Index | | Cost of Debt Financing | |
|---|---|---|---|---|
| | **(1)** | **(2)** | **(3)** | **(4)** |
| $Treat_i \times Post_t$ | −0.214 *** | −0.097 ** | −0.012 *** | −0.010 *** |
| | (−4.66) | (−2.56) | (−7.62) | (−6.67) |
| $ln$age | | 1.189 *** | | 0.056 *** |
| | | (9.43) | | (11.47) |
| soe | | 0.121 | | −0.006 * |
| | | (1.36) | | (−1.68) |
| $ln$size | | −0.704 *** | | −0.009 *** |
| | | (−22.34) | | (−7.63) |
| lev | | 7.627 *** | | 0.143 *** |
| | | (61.94) | | (29.73) |
| $ln$top1 | | −0.289 *** | | −0.011 *** |
| | | (−5.13) | | (−4.84) |
| roa | | −5.414 *** | | 0.018 ** |
| | | (−27.23) | | (2.26) |
| Constant | 1.124 *** | −1.519 *** | 0.008 *** | −0.134 *** |
| | (56.44) | (−3.49) | (12.30) | (−7.87) |
| Control variables | Yes | Yes | Yes | Yes |
| Year FE | Yes | Yes | Yes | Yes |
| Industry FE | Yes | Yes | Yes | Yes |
| Id FE | Yes | Yes | Yes | Yes |
| Observations | 16,462 | 16,462 | 17,084 | 17,084 |
| $R^2$ | 0.604 | 0.732 | 0.433 | 0.477 |

Note: ***, **, and * indicate statistical significance at 1%, 5%, and 10%, respectively.

The regression results in columns (1) and (3) in Table 5 show that coefficient $\beta_1$ estimates are both negative at the 1% significance level. This shows that NTU might encourage the development of green technologies by eliminating enterprise financing restrictions and lowering the cost of enterprise debt financing to encourage enterprises to increase their R&D spending. The possible reasons for this are that green financial policies in pilot regions not only target green and low-carbon enterprises, but also, their general credit policies for enterprises other than green industries reflect some green technological innovation effects.

In terms of control variables, overall, the policy is more effective in reducing the financing constraints for enterprises with large-sized assets, a higher equity concentration, and a higher return on assets. At this point, Hypothesis H3 has been verified.

### 5.4. Heterogeneity Analysis

5.4.1. Heterogeneity of Enterprise Regional Distribution

To determine whether NTU has diverse effects on green technology innovation by enterprises in various regions, according to national policy divisions, the sample is divided into eastern, central, and western enterprises in this paper, which also analyzes regional heterogeneity. After setting the geographical dummy variables, I obtained the precise regression outcomes displayed in Table 6.

When three fixed effects are simultaneously controlled for, coefficient $\beta_1$ is considerably positive in columns (2) and (3), relating to the central and western region's enterprises, and it is negligible in column (1), pertaining to the eastern regions' enterprises.

This illustrates that the impact of NTU on enterprises' green technology innovation is indeed heterogeneous at different levels of the regions of the enterprises and that enterprises in central and western regions may considerably benefit from the policy's promotion of green technology innovation, while the effect is not significant for enterprises in eastern regions.

**Table 6.** Enterprise heterogeneity analysis results.

| Variables | East Region | Middle Region | West Region | Highly Marketable Areas | Less Marketable Areas | Heavy-Polluting Enterprises | Non-Heavily-Polluting Enterprises | High-Innovation-Capability Enterprise | Low-Innovation-Capacity Enterprises |
|---|---|---|---|---|---|---|---|---|---|
| | (1) | (2) | (3) | (4) | (5) | (6) | (7) | (8) | (9) |
| $Treat_i \times Post_t$ | 0.014 | 0.173 *** | 0.118 *** | 0.053 * | −0.013 | 0.051 | 0.059 *** | 0.074 ** | 0.011 |
| | (0.67) | (4.05) | (2.82) | (1.74) | (−0.56) | (1.46) | (3.04) | (2.04) | (1.54) |
| $ln$age | 0.103 | 0.068 | −0.509 *** | 0.010 | 0.279 *** | −0.078 | 0.135 ** | −0.108 | 0.040 * |
| | (1.54) | (0.48) | (−2.84) | (0.09) | (3.43) | (−0.57) | (2.15) | (−0.88) | (1.68) |
| soe | 0.071 | 0.306 *** | 0.128 | 0.077 | 0.140 ** | 0.051 | 0.124 *** | 0.034 | 0.051 *** |
| | (1.39) | (3.53) | (1.20) | (1.28) | (2.26) | (0.59) | (2.71) | (0.45) | (2.79) |
| $ln$size | 0.325 *** | 0.357 *** | 0.282 *** | 0.358 *** | 0.238 *** | 0.319 *** | 0.320 *** | 0.382 *** | 0.024 *** |
| | (18.66) | (10.29) | (7.67) | (14.96) | (11.43) | (9.32) | (20.20) | (11.99) | (3.96) |
| lev | −0.26 1*** | −0.255 * | −0.002 | −0.166 * | −0.277 *** | −0.166 | −0.201 *** | −0.353 *** | −0.044 * |
| | (−3.88) | (−1.86) | (−0.02) | (−1.81) | (−3.54) | (−1.34) | (−3.24) | (−2.91) | (−1.95) |
| $ln$top1 | −0.026 | −0.003 | −0.240 *** | −0.057 | −0.021 | −0.017 | −0.058 ** | 0.018 | −0.023 ** |
| | (−0.85) | (−0.05) | (−3.40) | (−1.39) | (−0.54) | (−0.31) | (−2.01) | (0.32) | (−2.19) |
| roa | −0.168 | −0.205 | −0.162 | −0.248 ** | −0.057 | −0.068 | −0.191 * | −0.231 | 0.053 |
| | (−1.56) | (−0.88) | (−0.69) | (−1.99) | (−0.42) | (−0.33) | (−1.91) | (−1.20) | (1.46) |
| Constant | −0.705 *** | −1.159 ** | 1.537 *** | −0.551 | −0.974 *** | −0.369 | −0.769 *** | 0.232 | −0.069 |
| | (−3.03) | (−2.42) | (2.59) | (−1.50) | (−3.47) | (−0.84) | (−3.49) | (0.56) | (−0.84) |
| Control variables | Yes | Yes | Yes | Yes | Yes | Yes | Yes | Yes | Yes |
| Year FE | Yes | Yes | Yes | Yes | Yes | Yes | Yes | Yes | Yes |
| Industry FE | Yes | Yes | Yes | Yes | Yes | Yes | Yes | Yes | Yes |
| Id FE | Yes | Yes | Yes | Yes | Yes | Yes | Yes | Yes | Yes |
| Observations | 11,724 | 3013 | 2338 | 8483 | 8527 | 3634 | 13,380 | 6454 | 10,309 |
| $R^2$ | 0.744 | 0.689 | 0.705 | 0.746 | 0.789 | 0.721 | 0.735 | 0.735 | 0.253 |

Note: ***, **, and * indicate statistical significance at 1%, 5%, and 10%, respectively.

The possible reasons for this are: (1) Due to objective differences in the governance capacities between regional governments, the level of practice in pilot cities in the east has stabilized in the exploration of the development model of the region, while central and western regions are better able to take advantage of NTU to guide enterprises in green technology innovation. (2) Compared with the eastern region, in the western and central cities that have adopted a significant number of pollution-intensive enterprises during industrialization processes, the issues of high pollution levels and high-level energy consumption are particularly pronounced. Under the pressure of NTU policies, regional governments are more inclined to guide enterprises to transform their green patent achievements into economic and ecological benefits and achieve transformation, considering the difficulty of innovation in different parts of the value chain. At this point, Hypothesis H2a has been verified.

5.4.2. Heterogeneity Analysis of Degree of Marketization

According to the different degrees of marketization in different regions in China, the research results of Fan Gang's marketization index, the data from "China's regional marketization process 2021 annual report", and the disclosed median level of regional marketization, I divided the sample into high-marketization and low-marketization regions, and the results are shown in Table 6. Column (5) corresponds to enterprises in the low-marketization area, and its coefficient is not significant. Column (5)'s 10% level coefficient is significant and positive, corresponding to the high degree of market-oriented enterprises in the region. This shows that NTU has a strong role in promoting the green technology innovation of enterprises in regions with a high marketization degree. The possible reasons are: (1) the enterprises in regions that are marketable need to strengthen the green technology innovation because of competitive pressure, so as to reduce the cost of pollution reduction and obtain a higher emission reduction income, thus cultivating the competitive advantage in the market; (2) enterprises in regions that are not marketable may have less competitive pressure and a low level of enthusiasm for green technology innovation, so the momentum of green technology innovation is weak. So far, the Hypothesis H2b has been validated.

### 5.4.3. Heterogeneity Analysis of Industrial Pollution Degree

Based on the Environmental Information Disclosure Guide for listed companies issued by the Ministry of Ecology and Environment in 2012, I divided the research samples into heavy-polluting industry enterprises and non-heavy-polluting industry enterprises, placing the heavy-polluting industries into eight categories: metals and non-metals, water, electricity and gas, extractive industries, petrochemical plastics, food and beverage, textiles and furs, paper and printing, and biomedicine; the other industries are non-heavy-polluting industries. The results are shown in Table 6. Column (6) corresponds to enterprises in heavy-polluting industries, and its coefficient is not significant. Column (7)'s the 10% level coefficient is significant and positive, corresponding to enterprises in non-heavy-polluting industries. This shows that NTU plays a strong role in promoting green technology innovation in non-heavy-polluting industries. The possible reasons are as follows: (1) enterprises in non-heavy-polluting industries are in line with the value orientation of the NTU policy and have a greater possibility to obtain external resources to support green technology innovation; (2) enterprises in heavy-polluting industries occupy more R&D resources due to pollution control problems in the production process, which leads to an insufficient level of green technology innovation. So far, H2c has been validated.

### 5.4.4. Heterogeneity of Enterprise Innovation Capabilities

I separated the research sample into two subsamples: low-innovation-capability enterprises and high-innovation-capability enterprises based on the median of the total number of green patent applications during the preceding three years. The results are displayed in Table 6. Column (9) corresponding to low-innovation-capability enterprises; coefficient $\beta_1$ is not significant. Coefficient $\beta_1$ is significant and positive at the 1% level in column (8), corresponding to enterprises with a high innovation capability. This indicates that NTU has a stronger promotion effect on the green technology innovation of high-innovation-capability enterprises.

Possible reasons for this are: (1) The areas in which high-innovation-capability enterprises are located cause them to face greater competitive pressure in the market, emphasize efficiency and profit maximization, and obey the market logic more. After the government promulgates the pilot policy of NTU, high-innovation-capability enterprises will respond more quickly and actively to maintain their competitive advantages in the market. (2) Under the pressure of the NTU policy, regional governments are more inclined to invest in high-innovation-capability enterprises in order to accelerate the efficiency of green technology innovation and raise the level of local green achievement. So far, Hypothesis H2d has been verified.

## 6. Conclusions and Countermeasures

Based on the green patent application data of 1717 listed companies in Shanghai and Shenzhen A-shares in China from 2011 to 2020, this paper examined the impact effect, heterogeneity analysis, and transmission mechanism of NTU on enterprise innovation in green technology using the difference-in-differences model. Based on the findings of the experiments, before offering any policy suggestions, this section develops and discusses the major conclusions.

### 6.1. Conclusions

First, NTU could help enterprises with innovation in green technology to some extent. Even if the impact is reduced when time, the industry, and enterprises are considered, innovation in green technology is still significant. This result is true even after conducting tests such as the parallel trend test, the placebo test, and the PSM-DID test. This result verifies that the policy effect of NTU on enterprises' green technology innovation exists, which is consistent with H1. Although NTU is different from formal environmental regulation, it does not directly regulate the green development behavior of enterprises through legislation, but as a "soft constraint" environmental regulation, its direct impact

on enterprises' green technology innovation is consistent with other research conclusions on formal environmental regulation, which plays a positive role. On the one hand, from an institutional point of view, it has a certain binding effect on the green development of enterprises; on the other hand, the value orientation of NTU includes sustainable development. For enterprises, the environmental costs of enterprises are offset by the compensation effect of innovation; it is in line with the Porter hypothesis to provide conditions for enterprises' sustainable green technology innovation. Therefore, we need to pay attention to the positive role of NTU on enterprises' green technology innovation.

Second, heterogeneity analysis revealed that in the middle and western regions compared to the east, NTU has a higher impact on innovation in green technology. When it comes to fostering innovation in green technology, NTU is more beneficial to enterprises with a strong innovation capacity than it is to those with a poor innovation ability. Compared with enterprises in less marketable areas, NTU is more helpful in promoting enterprises in highly marketable areas to innovate in green technologies; compared with enterprises in heavy-pollution industries, NTU is more effective in promoting enterprises in highly marketable areas to innovate in green technologies, and NTU is more helpful in promoting green technology innovation in non-heavy-polluting industries; NTU is more helpful in promoting green technology innovation in highly innovative enterprises than it is in less innovative enterprises. These results confirm the research hypothesis and mean that NTU has a positive overall impact on green innovation, but it is necessary to more precisely identify the role of NTU in different contexts, the differences between eastern, middle and western regions, the degree of marketization, the degree of industry pollution, and how the green innovation ability can form the different conditions of NTU and Enterprises' green technology innovation.

Third, the mechanism analysis shows that NTU can enable enterprises to boost their R&D spending by easing their financial limitations and reducing their debt financing costs, thus promoting their green technology innovation. It indicates that green financial policies implemented in pilot areas not only target green and low-carbon enterprises, but also, their general credit policies reflect some green technological innovation effects for enterprises other than those of green industries. We focus on the role of NTU policy as an informal environmental regulator, as well as on supporting urban planning. NTU can provide subsidies, loans, special funds, and other green finance policies; from the perspective of resources, we can relax the financing constraint and promote the green technology innovation of enterprises.

Even if this work makes some contributions to the field, there are still issues that require fixing. First, this paper conducts a preliminary assessment of how the NTU policy has affected enterprises' adoption of innovative green technologies, but the interrelationship between the two needs further research in the future. Second, this paper does not individually analyze the impact of the NTU policy on enterprises by industry or ownership; instead, it solely examines the effects of the NTU policy on green technology innovation at the enterprise level overall. Third, due to data availability limitations, dynamic effects were not used to evaluate whether the NTU policy's impact on the development of green technology is delayed, which requires a longer sample period of enterprise green patent data for subsequent verification. At the same time, because the research data of NTU policy are mainly described by virtual variables, there is no suitable index of policy intensity to observe the implementation intensity of NTU; therefore, it is necessary to further explore the index construction of NTU policy implementation intensity in the future.

### 6.2. Countermeasures

First, the governments of pilot cities should design more reasonable, flexible, and market-oriented policy tools, including environmental taxes, emission fees, environmental subsidies, and emissions trading, to establish a reasonable price mechanism. By introducing more preferential policies, such as policy incentives and financial subsidies, thus easing the financing constraints of enterprises and reducing their debt financing costs,

the government can effectively improve the enthusiasm of enterprises to carry out green technology innovation.

Secondly, we need to accurately identify the impact of NTU policy on enterprises' green technology innovation under different heterogeneity conditions in order to encourage the market-oriented application of green innovation research; the governments of pilot cities should actively encourage the establishment of a green technology innovation system, with enterprises as the main body, and the deep integration of industry, academia, and research and accelerate the reform of the ownership of scientific and technological achievements by gradually improving the mechanism for the transformation of scientific and technological achievements.

Finally, due to the vast territory in China, the level of economic development and the level of government management vary greatly from region to region. When one is weighing the overall benefits of energy conservation and emission reduction of enterprises nationwide, the government needs to implement differentiated environmental regulatory policies according to the local conditions and provide special financial support for green technological innovation to underdeveloped regions in the central and western regions and enterprises lacking an innovation capacity, so as to prevent industrial transfer from aggravating pollution in the central and western regions and make the developments more balanced.

**Author Contributions:** R.Z.: Methodology, Formal analysis and investigation, Writing—Reviewing and Editing, Project administration, Funding acquisition. G.K.: Conceptualization, Software, Resources, Data curation, Writing—Original draft preparation. H.S.: Validation, Visualization, Supervision. All authors have read and agreed to the published version of the manuscript.

**Funding:** This research was funded by the National Social Science Fund of China (Grant no. 21BGL269).

**Institutional Review Board Statement:** Not applicable.

**Informed Consent Statement:** Not applicable.

**Data Availability Statement:** Not applicable.

**Conflicts of Interest:** The author declares no conflict of interest.

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
