# Peer review of "Can New-Type Urbanization Promote Enterprise Green Technology Innovation?—A Study Based on Difference-in-Differences Model"

_sustainability, doi:10.3390/su15076147_

Round 1

Reviewer 1 Report

The paper “Can new-type urbanization promote enterprise green technology innovation?—Based on difference-in-differences model” is interesting and has good policy implications. The overall structure and logic of the paper are reasonable, and the research topic is also in line with the journal’s requirements. But some problems need to be resolved before publication.

1. The descriptive statistical characteristics of the main variables should be provided with the comparison between the treated and control groups.

2. In the placebo test, there are usually two ways to test the robustness of the results. One is randomly choosing the policy-implementing period, and examining the timing of the policy is not an accident. The other is randomly selecting the treated and control groups to test other potential confounders that may affect the results. But you only provide one placebo test result, which should be explained the reason or add relevant results.

3. What puzzles me most is that China’s new-type urbanization goal is very complicated. Why does green technology innovation by enterprises have such a significant impact? The author should elaborate on the content related to new-type urbanization and green development and theoretically explain that the potential relationship between green technology innovation of listed enterprises is reasonable.

4. Are there any heterogeneous results for different kinds of listed enterprises? For example, are there differences in the sensitivity of listed enterprises in different industries to the policy?

5. If the intensity of policy implementation is different across regions, the degree of green technology innovation of listed enterprises may also be different. I hope the authors can provide the impact differences by the intensity of policy implementation.

6. The operating status of listed enterprises is usually related to the degree of marketization in a region, so the influence of market conditions can be further considered.

7. The comparison of the results with other related studies should be provided in the Discussion section.

Reviewer 2 Report

Dear/s Autohor/s,

The aim of the paper is to analyze a new-type urbanization (NTU) strategy in 2012 to solve the ecological and environmental problems brought by the traditional rapid and rough urbanization and environmental problems brought by the traditional rapid and rough urbanization development
model, making it a relevant and topical issue, given the importance that is being given worldwide to the concept of sustainable growth. The introduction and the review of the literature state the hypotheses well. The quantitative methodology used should be explained more clearly and appear in some form in the abastract. It is also observed that the conclusion seems more like a summary of the results and that there is no discussion section, where the results are contrasted with the theoretical framework that generates the hypotheses. Therefore, some structural deficiencies are observed, which need to be improved by the authors.

Best regards.

Round 2

Reviewer 1 Report

The paper is well-revised and is suitable for publication.